# Novel *Botrytis* and *Cladosporium* Species Associated with Flower Diseases of Macadamia in Australia

**DOI:** 10.3390/jof7110898

**Published:** 2021-10-25

**Authors:** Kandeeparoopan Prasannath, Roger G. Shivas, Victor J. Galea, Olufemi A. Akinsanmi

**Affiliations:** 1Queensland Alliance for Agriculture & Food Innovation, The University of Queensland, Ecosciences Precinct, Dutton Park, QLD 4102, Australia; 2Centre for Crop Health, University of Southern Queensland, Toowoomba, QLD 4350, Australia; roger.shivas@usq.edu.au; 3School of Agriculture & Food Sciences, The University of Queensland, Gatton, QLD 4343, Australia; v.galea@uq.edu.au

**Keywords:** Botrytis blight, Cladosporium blight, fungal ecology, raceme blight, taxonomy, tree nut

## Abstract

Macadamia (*Macadamia integrifolia*) is endemic to eastern Australia and produces an edible nut that is widely cultivated in commercial orchards globally. A survey of fungi associated with the grey and green mold symptoms of macadamia flowers found mostly species of *Botrytis* (Sclerotiniaceae, Leotiomycetes) and *Cladosporium* (Cladosporiaceae, Dothideomycetes). These isolates included *B. cinerea*, *C. cladosporioides*, and unidentified isolates. Amongst the unidentified isolates, one novel species of *Botrytis* and three novel species of *Cladosporium* were delimited and characterized by molecular phylogenetic analyses. The new species are *Botrytis macadamiae*, *Cladosporium devikae*, *C. macadamiae*, and *C. proteacearum*.

## 1. Introduction

*Macadamia* species and hybrids (*M. integrifolia* × *M. tetraphylla*) are native to Australia and are now grown worldwide in tropical and subtropical regions for their nuts that have edible kernels [1]. The expansion of macadamia orchards into new regions has led to an increase in the number and severity of diseases caused by fungi and oomycetes [2,3,4]. Flower and fruit diseases reduce the nut set and can cause significant yield losses in commercial macadamia orchards [5,6,7]. A mature macadamia tree can produce more than 10,000 racemes (inflorescences) during the flowering period, with 100–300 flowers per raceme [8,9]. Fruit and flower diseases often cause poor pollination that can reduce the nut set by 99% [10]. Diverse fungal pathogens are associated with flower blights of macadamia including *Botrytis cinerea* [11], *Cladosporium cladosporioides* [12], *Neopestalotiopsis macadamiae*, and *Pestalotiopsis macadamiae* [7].

Under high humidity and moisture, *B. cinerea* causes grey mold (Botrytis blight) that covers infected macadamia flowers with mycelia (Figure 1a) [11]. Index Fungorum accepted 71 *Botrytis* species (http://www.indexfungorum.org accessed on 17 September 2021), most of which are important pathogens of a wide range of host plants, including the grapevine, tomato, strawberry, bulbous crops, and cut flowers, causing devastating diseases during the preharvest and postharvest stages [13]. Among them, *B. cinerea* is one of the most important plant pathogens with wide-reaching economic and scientific impacts [14,15]. Many new species of *Botrytis* have been proposed [16] since Staats et al. [17] used molecular phylogenies to recognize *Botrytis* spp.

The genus *Cladosporium* (Cladosporiaceae, Dothideomycetes) was introduced by Link [18] with *C. herbarum* (Pers.) Link as the type species. *Cladosporium cladosporioides* causes flower blight known as green mold (Cladosporium blight) that manifests as olive-grey-colored mycelial patches with abundant conidia on macadamia racemes that later become necrotic (Figure 1b) [12]. *Cladosporium* spp. include endophytes, pathogens, and saprobes, and have a worldwide distribution across a range of substrates [19,20,21,22,23]. *Cladosporium* spp. are well-known as plant pathogens [19,24,25,26], and some can cause animal and human diseases [27,28,29]. Some pathogenic isolates of *Cladosporium* may have been wrongly classified as saprophytes, emphasizing the importance of the phylogenetic relationships for the identification of specialized lineages and cryptic species [24,28,30]. Some common species, *C. cladosporioides*, *C. herbarum*, and *C. sphaerospermum*, represent species complexes that await resolution as new isolates are collected from diverse ecosystems and geographical regions [19]. For example, *C. polonicum* and *C. neapolitanum* were described from within the *C. cladosporioides* species complex based on isolates recovered from galled flowers formed by midges on several species of Lamiaceae in Poland and Italy [31]. A phylogenetic analysis based on informative protein-coding genes is essential for the identification of species within *Botrytis* and *Cladosporium* genera [17,31].

Macadamia is a recently domesticated tree nut crop, with only *B. cinerea* and *C. cladosporioides* in their respective genera, reported as flower blight pathogens [11,12]. However, several unidentified isolates of *Botrytis* and *Cladosporium* were obtained from macadamia racemes with grey and green mold symptoms. Therefore, this study was aimed to determine the identity of the species of *Botrytis* and *Cladosporium* that are associated with flower diseases of macadamia in Australia.

## 2. Materials and Methods

### 2.1. Sample Collection and Isolation

The isolates included in this study were obtained from macadamia racemes with symptoms of grey and green mold diseases (Table 1). Samples were collected from commercial macadamia orchards in Queensland and New South Wales, Australia in 2019 and 2020. The specimens were surface sterilized and incubated, as described by Akinsanmi et al. [7]. Monoconidial cultures of the isolates were established, as described by Akinsanmi et al. [32], and preserved in a sterile 15% glycerol solution at −80 °C. Living cultures of the isolates were deposited in the Queensland Plant Pathology Herbarium (BRIP), Dutton Park, Australia.

### 2.2. Macro- and Micro-Morphological Studies

Colony characteristics of cultures on a ½-potato dextrose agar (PDA; Difco Laboratories, Franklin Lakes, NJ, USA) medium were photographed after 14 d of incubation at 25 °C. The fungal morphology was recorded from colonies grown in the dark for 14 d at 25 °C on PDA. Fungal structures were examined in lactic acid on slide mounts under a Leica DM5500B compound microscope (Wetzlar, Germany) with Nomarski differential interference contrast illumination, and images were captured with a Leica DFC 500 camera. Measurements of at least 30 conidia and other fungal structures were taken at 1000× magnification. Novel species were registered in MycoBank [33].

### 2.3. DNA Extraction, PCR Amplification, and Sequencing

Genomic DNA was extracted from approx. 40 mg of mycelium from colonies grown on PDA for 14 d. The mycelium was homogenized using TissueLyser (Qiagen, Chadstone, Australia) for 2 min at 30 Hz, and DNA was extracted using the BioSprint 96 DNA Plant Kit on a robotic platform (Qiagen, Chadstone, Australia). The DNA concentration was determined with a BioDrop Duo spectrophotometer (BioDrop, Cambridge, UK) and adjusted to 10 ng µL^−1^. For *Botrytis* isolates, partial sequences of the glyceraldehyde 3-phosphate dehydrogenase (*G3PDH*) gene with primers G3PDHfor+ and G3PDHrev+ [17], DNA-dependent RNA polymerase subunit II (*RPB2*) gene with primers RPB2for+ and RPB2rev+ [17], and heat shock protein 60 (*HSP60*) gene with primers HSP60for+ and HSP60rev+ [17] were amplified. For *Cladosporium* isolates, amplification was carried out using primers ITS4 and ITS5 [34] for the internal transcribed spacer (ITS) region of rDNA, primers EF1-526F and EF1-1567R [35] for partial sequences of the translation elongation factor 1-alpha (*TEF1α*) gene, and primers ACT-512F and ACT-783R [36] for the actin (*ACT*) gene sequences. The DNA of each isolate served as the template for the PCR amplification. Each reaction was performed in a 25 μL reaction mixture containing 5 μL of 5 × reaction buffer (Bioline, Eveleigh, Australia), 1.5 μL of 25 mM MgCl_2_, 0.5 μL of 10 mM dNTPs, 1 μL each of 10 μM forward and reverse primers, 0.125 μL of MangoTaq DNA polymerase (5 U/µL; Bioline, Eveleigh, Australia), 13.875 μL of nuclease free water, and 2 μL of DNA template. Amplification was performed in a SuperCycler Thermal Cycler (Kyratec, Wembley, Australia) with initial denaturation at 95 °C for 2 min, followed by 35 cycles at 95 °C for 30 s, an annealing step at 55 °C for 30 s, and elongation at 72 °C for 1 min, with a final extension step at 72 °C for 5 min. The quality of PCR amplicons was checked on 1% agarose gel electrophoresis stained with GelRed (Biotium, Melbourne, Australia) under UV light by Molecular Imager GelDoc (Bio-Rad Laboratories Inc., Gladesville, Australia). The targeted PCR products were purified and sequenced in both directions at Macrogen Inc. (Seoul, Korea).

### 2.4. Phylogenetic Analyses

The DNA sequences were assembled in Geneious Prime v. 2021.0.3 (Biomatters Ltd., San Diego, CA, USA), manually trimmed, and aligned to produce consensus sequences for each locus. The consensus sequences generated in this study were deposited in GenBank (Table 2 and Table 3). The sequences were compared against the NCBI GenBank nucleotide database using BLASTn to determine the closest phylogenetic relatives. The sequences of the reference isolates of the *Botrytis* (Table 2) and *Cladosporium* (Table 3) species were retrieved from GenBank and aligned with the sequences generated from our isolates using MAFFT v.7.3.8.8 [37] in Geneious. Ambiguously aligned positions in each multiple alignment were excluded using Gblocks v. 0.91b [38]. The concatenated three-locus sequence dataset (*RPB2* + *HSP60* + *G3PDH*) of *Botrytis* consisted of 42 taxa, with the outgroup taxon *Sclerotinia sclerotiorum* 484 (Table 2). The combined ITS, *TEF1α*, and *ACT* sequences of isolates belonging to the *C. cladosporioides* species complex comprised 72 taxa, with the outgroup taxon *C. herbarum* CBS 121,621 (Table 3). The combined sequence datasets were manually improved with BioEdit v. 7.2.5 [39], and gaps were treated as missing data. Phylogenetic trees were generated from Maximum Likelihood (ML), Bayesian Inference (BI), and Maximum Parsimony (MP) analyses.

The ML analysis was implemented using RAxML v. 8.2.11 [40] in Geneious. The search option was set to rapid bootstrapping, and the analysis was run using the GTR + G + I substitution model with 1000 bootstrap iterations. The BI analysis was conducted with MrBayes v. 3.2.1 [41] in Geneious to calculate posterior probabilities by the Markov Chain Monte Carlo (MCMC) method. The GTR + G + I evolution model was applied in the BI analysis. Four MCMC chains were run simultaneously, starting from random trees for 1,000,000 generations. The temperature of the heated chain was set to 0.25, and trees were sampled every 200 generations until the average standard deviation of split frequencies reached 0.01 (stop value). Burn-in was set at 25%, after which the likelihood values were stationary. The MP analysis was performed with PAUP v. 4.0b10 [42]. Trees were inferred using a heuristic search strategy with a 100 random stepwise addition and tree-bisection-reconnection (TBR) branch swapping. Max-trees were set to 5000, and bootstrap support values were evaluated for tree branches with 1000 replications [43]. Phylograms were visualized in FigTree v. 1.4.4 [44] and annotated in Adobe Illustrator 2021.

## 3. Results

### 3.1. Phylogenetic Analyses

The concatenated sequence data matrix of *Botrytis* comprised 2950 base pairs (bp) (1093 for *RPB2*, 971 for *HSP60*, and 886 for *G3PDH*), of which 2240 bp were constant, 296 bp were parsimony uninformative, and 414 bp were parsimony informative. The ML analysis yielded a best scoring tree, with a final ML optimization value of −11,930.57 and the following model parameters: alpha–0.561, Π(A)–0.268, Π(C)–0.241, Π(G)–0.237, and Π(T)–0.254.

The combined sequence dataset of *Cladosporium* consisted of 1000 bp (494 for ITS, 297 for *TEF1α*, and 209 for *ACT*), of which 678 bp were constant, 73 bp were parsimony uninformative, and 249 bp were parsimony informative. The ML analysis resulted in a best scoring tree with a final ML optimization value of −10,089.02 and the following model parameters: alpha–0.675, Π(A)–0.212, Π(C)–0.311, Π(G)–0.250, and Π(T)–0.227.

The tree topology generated by the ML analysis was similar to that of the BI and MP analyses. The best scoring ML phylograms of *Botrytis* and *Cladosporium* are shown in Figure 2 and Figure 3, respectively. ML bootstrap values, BI posterior probabilities, and MP bootstrap values (MLBS/BIPP/MPBS) are given at nodes of the phylogenetic trees (Figure 2 and Figure 3). The phylogenetic tree inferred from the concatenated alignment resolved the four *Botrytis* isolates associated with the grey mold symptoms into an independent monophyletic clade with high support that represents a novel species within the *Botrytis* genus (Figure 2). The phylogram inferred from the combined sequence data assigned four *Cladosporium* isolates associated with the green mold symptoms into three well-supported monophyletic clades that represent novel species within the *Cladosporium* genus (Figure 3).

### 3.2. Taxonomy

***Botrytis macadamiae*** Prasannath, Akinsanmi & R.G. Shivas, sp. nov. (Figure 4).

MycoBank: MB841218.

**Etymology**: Named after *Macadamia*, from which the type was first isolated.

**Type**: AUSTRALIA, New South Wales, Knockrow, from flower blight of *Macadamia integrifolia*, 25 October 2019, *J. Coates* (**Holotype** BRIP 72295a, includes ex-type culture). GenBank: MZ344226 (*G3PDH*); MZ344237 (*HSP60*); MZ356233 (*RPB2*).

**Description:***Hyphae* hyaline to pale brown, septate, 3–8 μm wide. *Sclerotia* single, sparse, dark grey to black, irregular to spherical, immersed, scattered, 0.2–2 mm diam. *Conidiophores* branched at top, erect, septate, subhyaline to pale brown, 1020–2050 × 10–20 μm. *Conidiogenous* cells swollen at the apex, 10–12 × 12–14 μm. *Conidia* in botryose clusters, elliptical to ovoid, unicellular, hyaline to pale brown, 9–11 × 6–7.5 μm.

**Culture characteristics:** Colonies on PDA at 25 °C after 14 d cover the plate, pale grey, abundant aerial mycelium in dark grey irregular tufts that cover most of the surface; reverse pale grey to buff brown.

**Habitat and distribution:** Racemes of *Macadamia integrifolia* (Proteaceae); Australia.

**Other material examined:** AUSTRALIA, New South Wales, Alstonville, from flower blight of *Macadamia*
*integrifolia*, 17 August 2019, *K. Prasannath* (living cultures, BRIP 72259a and BRIP 72261a); AUSTRALIA, New South Wales, Fernleigh, from flower blight of *Macadamia*
*integrifolia*, 23 September 2019, *S. Hill* (living culture, BRIP 72276a).

**Notes:***Botrytis macadamiae* was placed in a strongly supported clade with *B. cinerea*, *B. eucalypti,* and *B. pelargonii*. BLASTn searches in GenBank showed that *B. macadamiae* (BRIP 72295a) differed from *B.*
*cinerea* (MUCL87) in *RPB2* (Identities 1070/1075, 0 gaps); from *B.*
*eucalypti* (CERC 7170) in *HSP60* (Identities 934/935, 0 gaps) and *RPB2* (Identities 1071/1075, 0 gaps); from *B.*
*pelargonii* (CBS497.50) in *RPB2* (Identities 1071/1075, 0 gaps).

***Cladosporium devikae*** Prasannath, Akinsanmi & R.G. Shivas, sp. nov. (Figure 5).

MycoBank: MB841219.

**Etymology**: Named after Devika Malkanthi De Costa, for her guidance and mentorship to the senior author.

**Type:** AUSTRALIA, New South Wales, Fernleigh, from flower blight of *Macadamia*
*integrifolia*, 23 Sep. 2019, *S. Hill* (**Holotype** BRIP 72278a, includes ex-type culture). GenBank: MZ303808 (ITS); MZ344193 (*TEF1α*); MZ344212 (*ACT*).

**Description:***Mycelium* composed of branched, septate, smooth to finely roughened, brown, 3–4.5 μm diam. hyphae. *Conidiophores* erect, flexuous, subcylindrical, branched and unbranched, 200–700 × 2.5–4 μm, multiseptate, giving rise to an apical conidiogenous apparatus with chains of branched conidia. *Primary ramoconidia* subcylindrical, 11–30 × 3–5 μm, pale brown, smooth to finely roughened, 0–1-septate; hila thickened, darkened, refractive, 1.5–3.0 µm diam. *Secondary ramoconidia* subcylindrical to fusoid to ellipsoidal, 5–11 × 2–4 μm, pale brown, smooth to finely roughened, aseptate; hila thickened, darkened, refractive, 0.5–1.5 µm diam. *Intercalary* and *terminal conidia* in branched chains (−10), ellipsoidal, 3.5–7 × 2–3 μm, subhyaline to pale brown, smooth, aseptate; hila thickened, darkened, refractive, 0.5 μm diam.

**Culture characteristics:** Colonies on PDA 70 mm diam. after 14 d at 25 °C, flat, olivaceous, with sparse aerial mycelium, margins even and smooth; reverse black.

**Habitat and distribution:** Racemes of *Macadamia*
*integrifolia* (Proteaceae); Australia.

**Notes:** *Cladosporium devikae* belongs to the *C. cladosporioides* species complex. *Cladosporium devikae* was a sister species to *C. anthropophilum* in the phylogeny. BLASTn searches in GenBank showed that *C. devikae* (BRIP 72278a) differed from *C. anthropophilum* ex-type (CBS 140685) in *ACT* (Identities 189/198, 0 gaps) and *TEF1α* (Identities 208/217, 1 gap).

***Cladosporium macadamiae*** Prasannath, Akinsanmi & R.G. Shivas, sp. nov. (Figure 6).

MycoBank: MB841220.

**Etymology**: Named after *Macadamia,* from which the type was first isolated.

**Type:** AUSTRALIA, Queensland, Nambour, from flower blight of *Macadamia*
*integrifolia*, 22 Aug. 2019, *O.A. Akinsanmi* (**Holotype** BRIP 72269a, includes ex-type culture). GenBank: MZ303810 (ITS); MZ344195 (*TEF1α*); MZ344214 (*ACT*).

**Description:** *Mycelium* composed of branched, septate, smooth to finely roughened, brown, 3–4.5 μm diam. hyphae. *Conidiophores* erect, flexuous, subcylindrical, branched and unbranched, 200–500 × 2.5–5 μm, pale brown, multiseptate, giving rise to an apical conidiogenous apparatus with chains of branched conidia. *Primary ramoconidia* subcylindrical, 15–30 × 3–5 μm, pale brown, smooth, 0–1-septate; hila thickened, darkened, refractive, 1.5–3 µm diam. *Secondary ramoconidia* subcylindrical to fusoid to ellipsoidal, 7–18 × 3–4 μm, pale brown, smooth, aseptate; hila thickened, darkened, refractive, 0.5–1.5 µm diam. *Intercalary* and *terminal conidia* in branched chains (−10), ellipsoidal, 3–7 × 2–3 μm, subhyaline to pale brown, smooth, aseptate; hila thickened, darkened, refractive, 0.5 μm diam.

**Culture characteristics:** Colonies on PDA 70 mm diam. after 14 d at 25 °C, flat, olivaceous, with sparse aerial mycelium, margins even and smooth; reverse black.

**Habitat and distribution:** Racemes of *Macadamia*
*integrifolia* (Proteaceae); Australia.

**Other material examined:** AUSTRALIA, Queensland, Maleny, from flower blight of *Macadamia*
*integrifolia*, 20 September 2019, *O.A. Akinsanmi* (living culture, BRIP 72287a).

**Notes:** *Cladosporium macadamiae* belongs to the *C. cladosporioides* species complex. BLASTn searches in GenBank showed that *C. macadamiae* (BRIP 72269a) differed from *C.*
*crousii* ex-type (CBS 140686) in *ACT* (Identities 199/209, 0 gaps) and *TEF1α* (Identities 182/213, 2 gaps); from *C.*
*endoviticola* ex-type (JZB390018) in *ACT* (Identities 153/170, 3 gaps) and *TEF1α* (Identities 248/266, 1 gap); from *C.*
*pseudocladosporioides* ex-type (CBS 125993) in *ACT* (Identities 197/209, 0 gaps) and *TEF1α* (Identities 279/293, 0 gaps).

***Cladosporium proteacearum*** Prasannath, Akinsanmi & R.G. Shivas, sp. nov. (Figure 7).

MycoBank: MB841221.

**Etymology**: Named after Poteaceae, the family of the host plant from which the type was first isolated.

**Type:** AUSTRALIA, New South Wales, Rosebank, from flower blight of *Macadamia*
*integrifolia*, 16 Oct. 2019, *P. Fraser* (**Holotype** BRIP 72301a, includes ex-type culture). GenBank: MZ303809 (ITS); MZ344194 (*TEF1α*); MZ344213 (*ACT*).

**Description:** *Mycelium* composed of branched, septate, smooth to finely roughened, brown, 3–4.5 μm diam. hyphae. *Conidiophores* erect, flexuous, subcylindrical, branched and unbranched, 150–500 × 2.5–4 μm, multiseptate, giving rise to an apical conidiogenous apparatus with chains of branched conidia. *Primary ramoconidia* subcylindrical, 12–48 × 3–5 μm, pale brown, smooth, 0–1-septate; hila thickened, darkened, refractive, 1.5–3 µm diam. *Secondary ramoconidia* subcylindrical to fusoid to ellipsoidal, 5–10 × 3–4 μm, pale brown, smooth, aseptate; hila thickened, darkened, refractive, 0.5–1.5 µm diam. *Intercalary* and *terminal conidia* in branched chains (−10), ellipsoidal, 4–5 × 2–3 μm, subhyaline to pale brown, smooth, aseptate; hila thickened, darkened, refractive, 0.5 μm diam.

**Culture characteristics:** Colonies on PDA 70 mm diam. after 14 d at 25 °C, flat, olivaceous, with sparse aerial mycelium, margins even and smooth; reverse olivaceous.

**Habitat and distribution:** Racemes of *Macadamia*
*integrifolia* (Proteaceae); Australia.

**Notes:** *Cladosporium proteacearum* belongs to the *C. cladosporioides* species complex. *Cladosporium proteacearum* was a sister to *C. cucumerinum* in a well-supported clade. BLASTn searches in GenBank showed that *C. proteacearum* (BRIP 72301a) differed from *C. cucumerinum* ex-type (CBS 171.52) in *ACT* (Identities 198/209, 0 gaps); ITS (Identities 481/494, 1 gap); *TEF1α* (Identities 274/297, 3 gaps).

## 4. Discussion

*Botrytis macadamiae*, *Cladosporium devikae, C. macadamiae,* and *C. proteacearum* were isolated from macadamia inflorescences with grey and green mold symptoms and subsequently described. Each species formed a well-supported monophyletic clade in the phylogenetic analysis. The ITS region of the nuclear rDNA discriminates *Botrytis* from other genera in Sclerotiniaceae, although ITS is not useful for the delineation of the *Botrytis* species [45]. The three nuclear protein-coding genes, *G3PDH*, *HSP60*, and *RPB2*, have been used to characterize the *Botrytis* species [17]. To date, 40 species are phylogenetically recognized in *Botrytis* [16,46], including *B. macadamiae.* Whether *B. macadamiae* causes grey mold in macadamia has yet to be ascertained.

Grey mold is the most common disease caused by the *Botrytis* species affecting different plant organs, including flowers, fruits, leaves, and shoots [47]. Vegetables and small fruit crops such as the tomato, raspberry, grape, strawberry, blueberry, apple, and pear are among the most severely affected by these pathogens [47]. The genus *Botrytis* consisting of necrotrophic species has a very broad host range (e.g., *B. cinerea* and *B. pseudocinerea*) impacting more than 1400 different plant species [13]. On the contrary, other species have a narrow host range or are even host-specific, including *B. fabae* (broad bean) and *B. calthae* (marsh marigold) [48]. In some circumstances, multiple *Botrytis* species co-infect the same host plant; e.g., *B. squamosa*, *B. allii*, and *B. aclada* all cause significant economic risk to commercial onion production [15]. Interestingly, *B. squamosa* is family-specific and pathogenic on the onion, garlic, and leek (*Allium* spp.), while the closely related sister species are restricted to the lily (*B. elliptica*) and daylily (*B. deweyae*) [49]. Diversity among the *Botrytis* species is shown by cultural characteristics, virulence, and host specificity. However, the unique feature among all grey mold fungi is their necrotrophic lifestyle in which they kill host cells via the secretion of effector proteins to induce cell death, obtain nutrients, and subsequently colonize dead plant tissue [49,50].

The *Cladosporium* species are known as common and abundant fungi in indoor and outdoor environments. The *Cladosporium* species are also economically important spoilage organisms of grains, fruits, and refrigerated meat [51,52,53]. Several *Cladosporium* species are pathogenic to a wide range of hosts [30]. Most *Cladosporium* species are saprobic, but some have also been reported as endophytes, phylloplane fungi, and hyperparasites on other fungi [54,55,56]. Certain species show the ability to produce compounds of medical interest or are relevant as potential biocontrol agents for plant diseases [57,58]. Some species are pathogens to various crops and can cause economically important diseases, while others have only endemic importance [59]. These fungi can cause diseases of plants, often with different names, depending on the infected plants and the type of symptoms. Pathogenic species of *Cladosporium* are known to cause leaf mold of the tomato [60] and scab disease on leaves of the cucumber, the strawberry, and tea [61,62,63]. *Cladosporium cladosporioides* has been reported as a pathogen of scab in papaya [64], sooty mold in the persimmon [65], blossom blight in the strawberry [66], and leaf spot in the tomato [67].

Three major species complexes are recognized within the genus *Cladosporium*, viz. the *C. cladosporioides, C. herbarum,* and *C. sphaerospermum* species complexes [30]. The species identification and delineation in *Cladosporium* require a multi-locus DNA sequence analysis of the ITS region of rDNA gene, partial *ACT*, and *TEF1α* gene sequences [30]. The molecular approach combined with morphological features allowed the recognition of more than 230 species within the genus *Cladosporium* [68]. Our phylogenetic analysis using these three loci placed *C. devikae, C. macadamiae,* and *C. proteacearum* in the *C. cladosporioides* species complex. These species were well-delineated from other species in the *C. cladosporioides* species complex.

The proper identification of species is essential for all biological studies. The present study found a high diversity of *Cladosporium* spp. on macadamia racemes with green mold symptoms. Future studies will determine whether *B. macadamiae, C. devikae, C. macadamiae,* and *C. proteacearum* are pathogens or saprobes on macadamia inflorescences. Living cultures of *B. macadamiae, C. devikae, C. macadamiae,* and *C. proteacearum* are preserved and accessible in BRIP as cryopreserved cultures for future research and study.

## 5. Conclusions

*Botrytis macadamiae*, *Cladosporium devikae, C. macadamiae,* and *C. proteacearum* were described and illustrated. These fungi were isolated from inflorescences of macadamia with grey and green mold symptoms in Australia. The pathogenicity of these novel species on macadamia racemes has yet to be examined. Cryopreserved isolates of these fungi are available in BRIP for future research.

## Figures and Tables

**Figure 1 jof-07-00898-f001:**
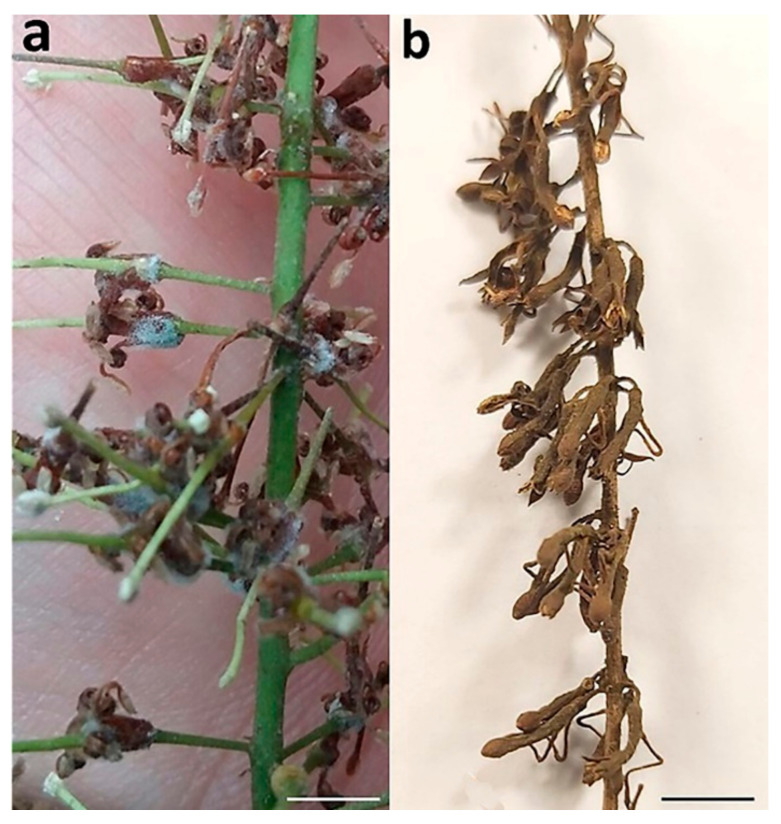
Macadamia racemes with symptoms of (**a**) grey mold, and (**b**) green mold. Scale bars: (**a**) = 5 mm; (**b**) = 10 mm.

**Figure 2 jof-07-00898-f002:**
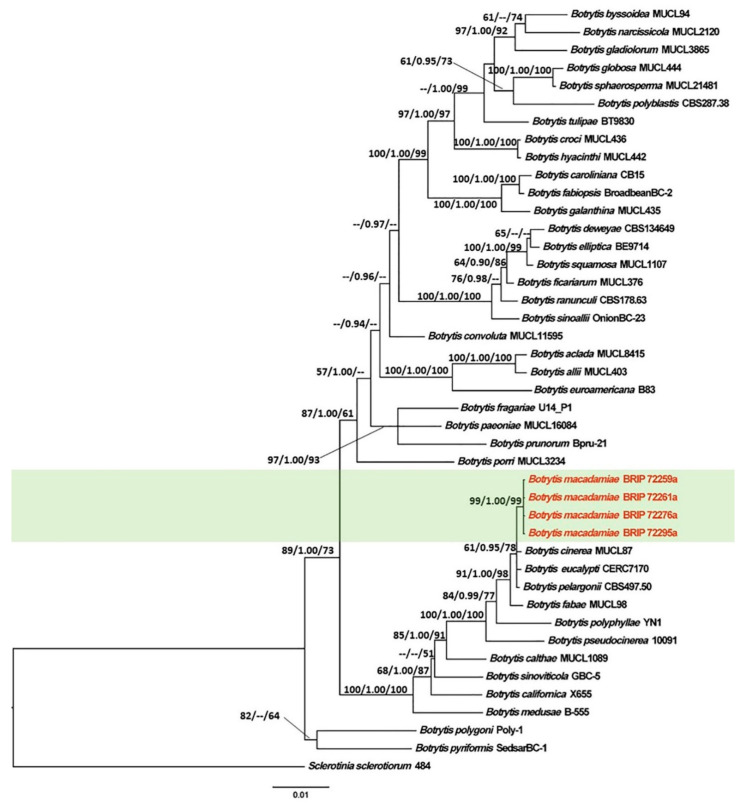
Maximum Likelihood tree topology of *Botrytis* based on a concatenated multi-locus alignment (*RPB2* + *HSP60* + *G3PDH*). *Sclerotinia sclerotiorum* 484 was used as an outgroup taxon. Maximum Likelihood bootstrap support values (>50%), Bayesian Inference posterior probabilities (>90%), and Maximum Parsimony bootstrap proportions (>50%) are displayed at the nodes, respectively. Isolates of the newly described species are depicted in red.

**Figure 3 jof-07-00898-f003:**
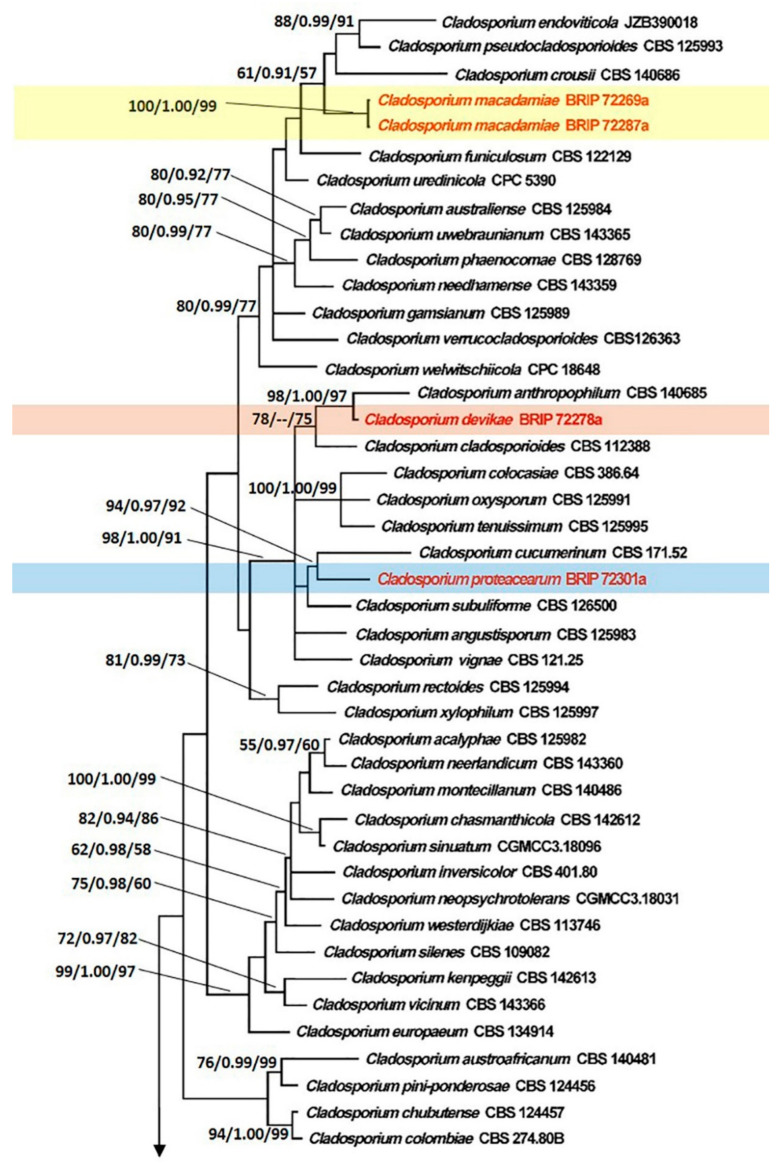
Maximum Likelihood tree topology of *Cladosporium* based on a combined multi-locus alignment (ITS + *TEF1α* + *ACT*). *Cladosporium herbarum* CBS 121621 was used as an outgroup taxon. Maximum Likelihood bootstrap support values (>50%), Bayesian Inference posterior probabilities (>90%), and Maximum Parsimony bootstrap proportions (>50%) are displayed at the nodes, respectively. Isolates of the newly described species are depicted in red.

**Figure 4 jof-07-00898-f004:**
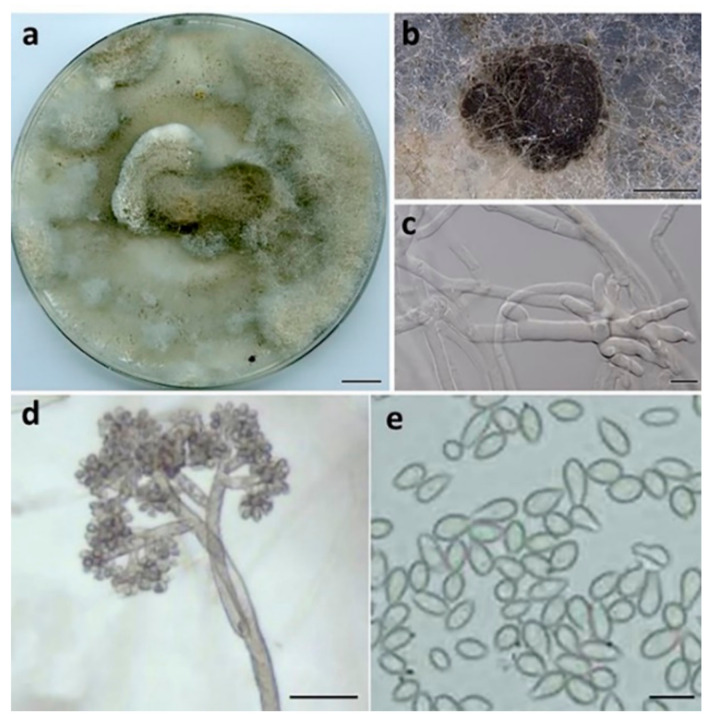
*Botrytis macadamiae* (BRIP 72295a). (**a**) Two-week-old colony on PDA, (**b**) sclerotia, (**c**) hyphae, (**d**) conidiophore, and (**e**) conidia. Scale bars: (**a**) = 1 cm; (**b**) = 1 mm; (**c**,**e**) = 10 µm; (**d**) = 50 µm.

**Figure 5 jof-07-00898-f005:**
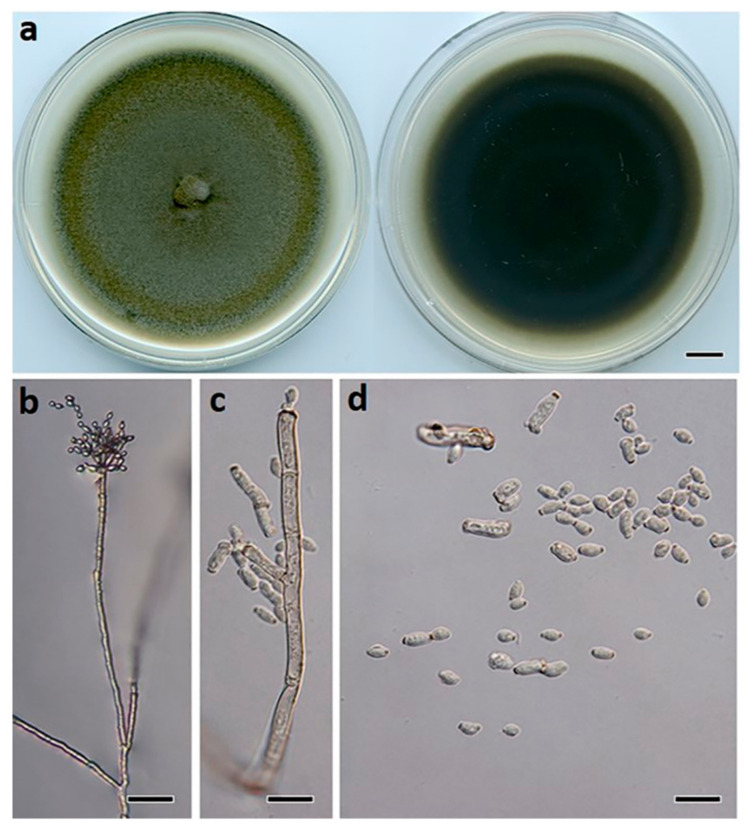
*Cladosporium devikae* (BRIP 72278a). (**a**) Two-week-old colony on PDA (upper surface and lower surface), (**b**) conidiophore, (**c**) ramoconidia, and (**d**) terminal conidia. Scale bars: (**a**) = 1 cm; (**b**) = 25 μm; (**c**,**d**) = 10 μm.

**Figure 6 jof-07-00898-f006:**
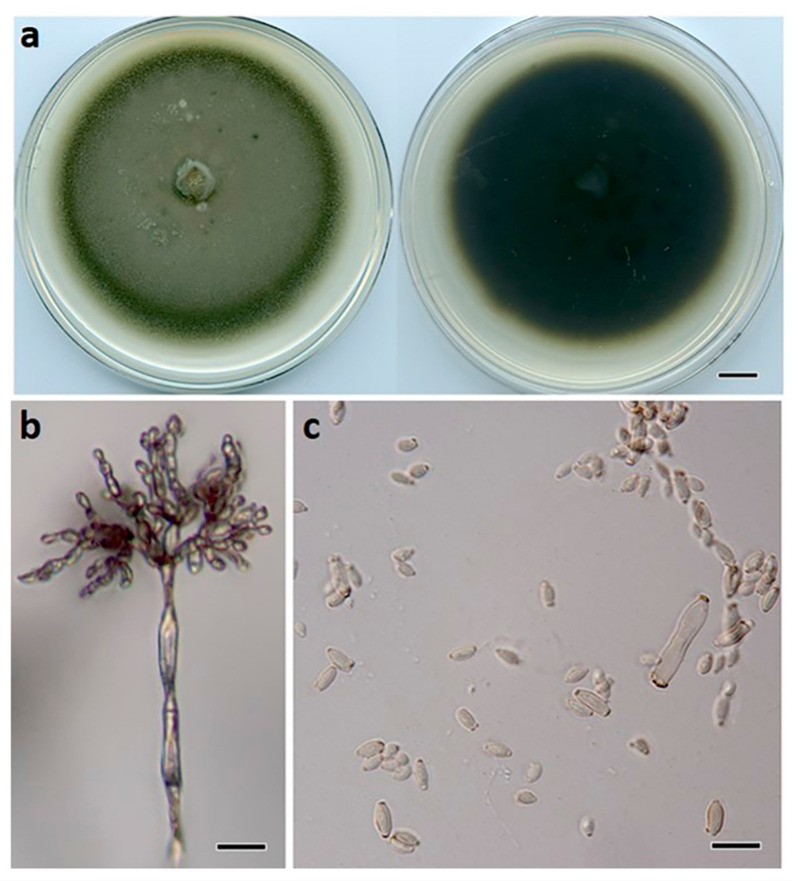
*Cladosporium macadamiae* (BRIP 72269a). (**a**) Two-week-old colony on PDA (upper surface and lower surface), (**b**) conidiophore, and (**c**) terminal conidia. Scale bars: (**a**) = 1 cm; (**b**) = 25 μm; (**c**) = 10 μm.

**Figure 7 jof-07-00898-f007:**
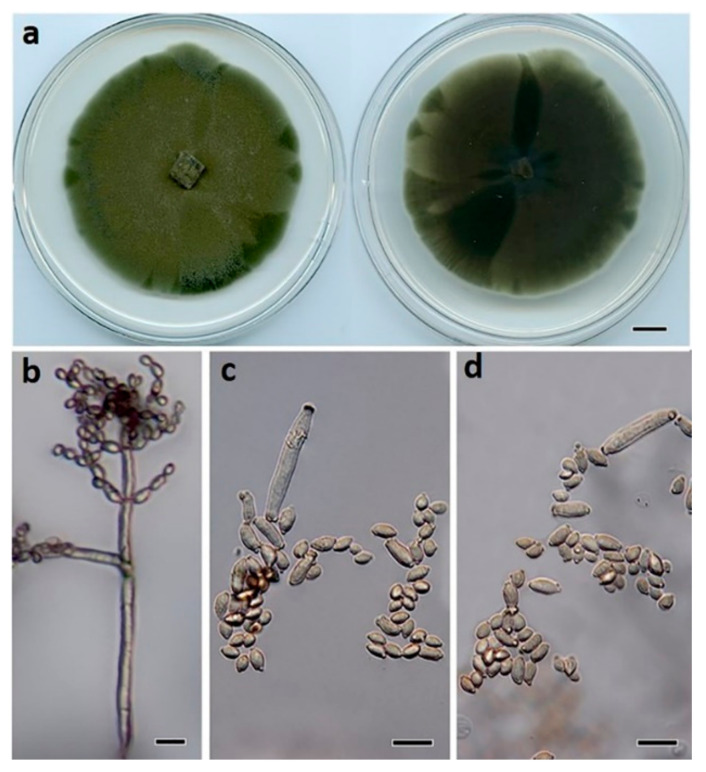
*Cladosporium proteacearum* (BRIP 72301a). (**a**) Two-week-old colony on PDA (upper surface and lower surface), (**b**) conidiophore, (**c**) ramoconidia, and (**d**) terminal conidia. Scale bars: (**a**) = 1 cm; (**b**) = 25 μm; (**c**,**d**) = 10 μm.

**Table 1 jof-07-00898-t001:** Details of *Botrytis* and *Cladosporium* isolates obtained from macadamia racemes with flower blight symptoms included in this study.

Isolate ^1^	Species	Cultivar	Flower Growth Stage	Location ^2^
BRIP 72259a	*Botrytis macadamiae*	HAES 246	3	Alstonville, NSW
BRIP 72261a	*B. macadamiae*	HAES 246	3	Alstonville, NSW
BRIP 72276a	*B. macadamiae*	HAES 344	3	Fernleigh, NSW
BRIP 72295a	*B. macadamiae*	A16	3	Knockrow, NSW
BRIP 72278a	*Cladosporium devikae*	HAES 344	1	Fernleigh, NSW
BRIP 72269a	*C. macadamiae*	HAES 792	4	Nambour, QLD
BRIP 72287a	*C. macadamiae*	A16	3	Maleny, QLD
BRIP 72301a	*C. proteacearum*	HAES 344	1	Rosebank, NSW

^1^ BRIP: Queensland Plant Pathology Herbarium (BRIP) accession numbers. ^2^ NSW: New South Wales, Australia; QLD: Queensland, Australia.

**Table 2 jof-07-00898-t002:** *Botrytis* species and isolates used in the phylogenetic analysis with GenBank accession numbers.

Species	Isolate	GenBank Accession Numbers ^1^		
		*G3PDH*	*HSP60*	*RPB2*
*Botrytis aclada*	MUCL8415	AJ704992	AJ716050	AJ745664
*B. allii*	MUCL403	AJ704996	AJ716055	AJ745666
*B. byssoidea*	MUCL94	AJ704998	AJ716059	AJ745670
*B. californica*	X655	KJ937069	KJ937059	KJ937049
*B. calthae*	MUCL1089	AJ705000	AJ716061	AJ745672
*B. caroliniana*	CB15	JF811584	JF811587	JF811590
*B. cinerea*	MUCL87	AJ705004	AJ716065	AJ745676
*B. convoluta*	MUCL11595	AJ705008	AJ716069	AJ745680
*B. croci*	MUCL436	AJ705009	AJ716070	AJ745681
*B. deweyae*	CBS 134649	HG799521	HG799519	HG799518
*B. elliptica*	BE9714	AJ705012	AJ716073	AJ745684
*B. eucalypti*	CERC 7170	KX301020	KX301024	KX301028
*B. euroamericana*	B83	KC191677	KC191678	KC191679
*B. fabae*	MUCL98	AJ705014	AJ716075	AJ745686
*B. fabiopsis*	BroadbeanBC–2	EU519211	EU514482	EU514473
*B. ficariarum*	MUCL376	AJ705016	AJ716077	AJ745688
*B. fragariae*	U14_P1	KX429699	KX429692	KX429706
*B. galanthina*	MUCL435	AJ705018	AJ716079	AJ745689
*B. gladiolorum*	MUCL3865	AJ705020	AJ716081	AJ745692
*B. globose*	MUCL444	AJ705022	AJ716083	AJ745693
*B. hyacinthi*	MUCL442	AJ705024	AJ716085	AJ745696
** *B. macadamiae* **	**BRIP 72259a**	**MZ344223**	**MZ344234**	**MZ356230**
	**BRIP 72261a**	**MZ344224**	**MZ344235**	**MZ356231**
	**BRIP 72276a**	**MZ344225**	**MZ344236**	**MZ356232**
	**BRIP 72295a**	**MZ344226**	**MZ344237**	**MZ356233**
*B. medusae*	B–555	MH732861	MH732866	MH732870
*B. narcissicola*	MUCL2120	AJ705026	AJ716087	AJ745697
*B. paeoniae*	MUCL16084	AJ705028	AJ716089	AJ745700
*B. pelargonii*	CBS497.50	AJ704990	AJ716046	AM087030
*B. polyblastis*	CBS287.38	AJ705030	AJ716091	AJ745702
*B. porri*	MUCL3234	AJ705032	AJ716093	AJ745704
*B. prunorum*	Bpru–21	KP339980	KP339994	KP339987
*B. pseudocinerea*	10091	JN692414	JN692400	JN692428
*B. pyriformis*	SedsarBC–1	KJ543484	KJ543488	KJ543492
*B. ranunculi*	CBS178.63	AJ705034	AJ716095	AJ745706
*B. sinoallii*	OnionBC–23	EU519217	EU514488	EU514479
*B. sinoviticola*	GBC–5	JN692413	JN692399	JN692427
*B. sphaerosperma*	MUCL21481	AJ705035	AJ716096	AJ745708
*B. squamosa*	MUCL1107	AJ705037	AJ716098	AJ745710
*B. tulipae*	BT9830	AJ705041	AJ716102	AJ745713
*Sclerotinia sclerotiorum*	484	AJ705044	AJ716048	AJ745716

^1^ G3PDH: glyceraldehyde 3-phosphate dehydrogenase; HSP60: Heat shock protein 60; RPB2: DNA-dependent RNA polymerase subunit II. The name and isolates of the new species, and newly generated sequences, are shown in bold font.

**Table 3 jof-07-00898-t003:** *Cladosporium* species and isolates used in the phylogenetic analysis with GenBank accession numbers.

Species	Isolate	GenBank Accession Numbers ^1^		
		ITS	*TEF1α*	*ACT*
*Cladosporium acalyphae*	CBS 125982 ^T^	HM147994	HM148235	HM148481
*C. alboflavescens*	CBS 140690 ^T^	LN834420	LN834516	LN834604
*C. angulosum*	CBS 140692 ^T^	LN834425	LN834521	LN834609
*C. angustisporum*	CBS 125983 ^T^	HM147995	HM148236	HM148482
*C. angustiterminale*	CBS 140480 ^T^	KT600379	KT600476	KT600575
*C. anthropophilum*	CBS 140685 ^T^	LN834437	LN834533	LN834621
*C. arenosum*	CHFC–EA 566	MN879328	MN890011	MN890008
*C. asperulatum*	CBS 126340 ^T^	HM147998	HM148239	HM148485
*C. australiense*	CBS 125984 ^T^	HM147999	HM148240	HM148486
*C. austroafricanum*	CBS 140481 ^T^	KT600381	KT600478	KT600577
*C. chalastosporoides*	CBS 125985 ^T^	HM148001	HM148242	HM148488
*C. chasmanthicola*	CBS 142612 ^T^	KY646221	KY646227	KY646224
*C. chubutense*	CBS 124457 ^T^	FJ936158	FJ936161	FJ936165
*C. cladosporioides*	CBS 112388 ^T^	HM148003	HM148244	HM148490
*C. colocasiae*	CBS 386.64 ^T^	HM148067	HM148310	HM148555
*C. colombiae*	CBS 274.80B ^T^	FJ936159	FJ936163	FJ936166
*C. crousii*	CBS 140686 ^T^	LN834431	LN834527	LN834615
*C. cucumerinum*	CBS 171.52 ^T^	HM148072	HM148316	HM148561
** *C. devikae* **	**BRIP 72278a ^T^**	**MZ303808**	**MZ344193**	**MZ344212**
*C. endoviticola*	JZB390018 ^T^	MN654960	MN984228	MN984220
*C. europaeum*	CBS 134914 ^T^	HM148056	HM148298	HM148543
*C. exasperatum*	CBS 125986 ^T^	HM148090	HM148334	HM148579
*C. exile*	CBS 125987 ^T^	HM148091	HM148335	HM148580
*C. flabelliforme*	CBS 126345 ^T^	HM148092	HM148336	HM148581
*C. flavovirens*	CBS 140462 ^T^	LN834440	LN834536	LN834624
*C. funiculosum*	CBS 122129 ^T^	HM148094	HM148338	HM148583
*C. gamsianum*	CBS 125989 ^T^	HM148095	HM148339	HM148584
*C. globisporum*	CBS 812.96 ^T^	HM148096	HM148340	HM148585
*C. grevilleae*	CBS 114271 ^T^	JF770450	JF770472	JF770473
*C. herbarum*	CBS 121621 ^T^	EF679363	EF679440	EF679516
*C. hillianum*	CBS 125988 ^T^	HM148097	HM148341	HM148586
*C. inversicolor*	CBS 401.80 ^T^	HM148101	HM148345	HM148590
*C. ipereniae*	CBS 140483 ^T^	KT600394	KT600491	KT600589
*C. iranicum*	CBS 126346 ^T^	HM148110	HM148354	HM148599
*C. kenpeggii*	CBS 142613 ^T^	KY646222	KY646228	KY646225
*C. licheniphilum*	CBS 125990 ^T^	HM148111	HM148355	HM148600
*C. longicatenatum*	CBS 140485 ^T^	KT600403	KT600500	KT600598
** *C. macadamiae* **	**BRIP 72269a ^T^**	**MZ303810**	**MZ344195**	**MZ344214**
	**BRIP 72287a**	**MZ303811**	**MZ344196**	**MZ344215**
*C. montecillanum*	CBS 140486 ^T^	KT600406	KT600504	KT600602
*C. myrtacaearum*	CBS 126350 ^T^	HM148117	HM148361	HM148606
*C. needhamense*	CBS 143359 ^T^	MF473142	MF473570	MF473991
*C. neerlandicum*	CBS 143360 ^T^	KP701887	KP701764	KP702010
*C. neopsychrotolerans*	CGMCC3.18031 ^T^	KX938383	KX938400	KX938366
*C. oxysporum*	CBS 125991	HM148118	HM148362	HM148607
*C. paracladosporioides*	CBS 171.54 ^T^	HM148120	HM148364	HM148609
*C. parapenidielloides*	CBS 140487 ^T^	KT600410	KT600508	KT600606
*C. perangustum*	CBS 125996 ^T^	HM148121	HM148365	HM148610
*C. phaenocomae*	CBS 128769 ^T^	JF499837	JF499875	JF499881
*C. phyllactiniicola*	CBS 126355 ^T^	HM148153	HM148397	HM148642
*C. phyllophilum*	CBS 125992 ^T^	HM148154	HM148398	HM148643
*C. pini-ponderosae*	CBS 124456 ^T^	FJ936160	FJ936164	FJ936167
** *C. proteacearum* **	**BRIP 72301a ^T^**	**MZ303809**	**MZ344194**	**MZ344213**
*C. pseudochalastosporoides*	CBS 140490 ^T^	KT600415	KT600513	KT600611
*C. pseudocladosporioides*	CBS 125993 ^T^	HM148158	HM148402	HM148647
*C. rectoides*	CBS 125994 ^T^	HM148193	HM148438	HM148683
*C. rugulovarians*	CBS 140495 ^T^	KT600459	KT600558	KT600656
*C. scabrellum*	CBS 126358 ^T^	HM148195	HM148440	HM148685
*C. silenes*	CBS 109082 ^T^	EF679354	EF679429	EF679506
*C. sinuatum*	CGMCC3.18096 ^T^	KX938385	KX938402	KX938368
*C. subuliforme*	CBS 126500 ^T^	HM148196	HM148441	HM148686
*C. tenuissimum*	CBS 125995 ^T^	HM148197	HM148442	HM148687
*C. tianshanense*	CGMCC3.18033 ^T^	KX938381	KX938398	KX938364
*C. uredinicola*	CPC 5390	AY251071	HM148467	HM148712
*C. uwebraunianum*	CBS 143365 ^T^	MF473306	MF473729	MF474156
*C. varians*	CBS 126362 ^T^	HM148224	HM148470	HM148715
*C. verrucocladosporioides*	CBS 126363 ^T^	HM148226	HM148472	HM148717
*C. vicinum*	CBS 143366 ^T^	MF473311	MF473734	MF474161
*C. vignae*	CBS 121.25	HM148227	HM148473	HM148718
*C. welwitschiicola*	CPC 18648 ^T^	KY646223	KY646229	KY646226
*C. westerdijkiae*	CBS 113746 ^T^	HM148061	HM148303	HM148548
*C. xanthocromaticum*	CBS 140691 ^T^	LN834415	LN834511	LN834599
*C. xylophilum*	CBS 125997 ^T^	HM148230	HM148476	HM148721

^1^ ITS: internal transcribed spacer; *TEF1α*: translation elongation factor 1-α; *ACT*: actin. ^T^ Ex-type isolates. The name and isolates of the new species, and newly generated sequences, are shown in bold font.

## Data Availability

All sequence data are available in NCBI GenBank (www.ncbi.nlm.nih.gov) following the accession numbers in the manuscript.

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
