# Peer review of "Novel Botrytis and Cladosporium Species Associated with Flower Diseases of Macadamia in Australia"

_jof, 2021, doi:10.3390/jof7110898_

Round 1

Reviewer 1 Report

Dear Authors,

in my opinion your work is very interesting in a cognitive context and contributes a lot to mycology, molecular and evolutionary taxonomy. All the tables and figures are appropriate for this type of article. In general, the paper has a logical flow and it is refined in detail. The abstract well correspond with the main aspects of the work. Nevertheless, I see one important shortcoming, namely a discussion in my opinion is too short as for a paper describing new species within the well-known Cladosporium and Botrytis genera. Especially in the context of the issues initiated in the last paragraph (lines 330-335), it is worth giving examples and characteristics of species which are pathogenic or saprotrophic amongst Cladosporium genus. I am convinced that the Authors are able to resolve this problem very fast.

As a reviewer I am obligated to pay attention even to less important weak points of this work and all mentioned below comments should be carefully considered.

Line 34

In my opinion ,,…B. cinerea causes gray-mould rot (Botrytis blight)…” sounds better and is correct, because B. cinerea is called ,,gray mould” but causes ,,Botrytis blight” or (other name) ,,gray-mould rot”

Line 59

,,…identification of species within Botrytis and Cladosporium genera…” sounds better

Line 75

Due to the fact that this study concerns only species from Botrytis and Cladosporium genera which form exogenously conidia should be used term ,,single-conidia cultures" (spores according to definition arise endogenously). I propose to replace this term in the whole work.

Line 79

In my opinion ,,macro- and microscopic studies” or ,,Macro- and micromorphological studies” sounds better

Lines 81 and 82

,,were photographed” sounds better than ,,were recorded”

Lines 81 and 82

Please, lets check whether the abbreviation used (14 d incubation) is correct

Line 94

I  think there should be ,, 10 ng x µL-1” of course with ,,-1” with upper script

Lines 89-114

All the gene names should be written in italic

Table 1

All the species names should be written in italic

Lines 176 and 179

In my opinion ,,…a novel species within Botrytis genus…” and ,,…a novel species within Cladosporium genus…” sounds better

Line 189

According to the best of my knowledge correct is ,,conidiophores septated”

Lines 238-240

In the context of ,,Intercalary and terminal conidia” please provide information about conidia, namely whether these are septated or aseptated

Line 245

In my opinion ,,Cladosporium devikae was sister species to C. anthropophilum in the phylogeny” sounds better

Figures 4-7

In the headings of figures 4-7, I propose to replace the terms ,,upper surface” and ,,lower surface” with ,,front” and ,,reverse”, respectively

Line 311

In my opinion something is wrong in this sentence ,,…C. proteacearum were discovered from the isolates obtained from macadamia inflorescences…”. For me sounds better if authors use ,,…C. proteacearum were isolated from macadamia inflorescences…”

Lines 316-317

Gene names should be written in italic. It is also worth checking the entire manuscript in this regard

Line 323

In my opinion ,,pathogenic to the wide range of hosts” sounds better

Reviewer 2 Report

This is a very well written paper resulting from very complete analyses. The paper could be improved by adding a table that gives the locations where the new isolates were collected (instead of having to read through every new species description) and abundance (how many times overall that each were isolated).
